# GLID$^2$E: Lightweight Policy-Based Fine-Tuning for Discrete Diffusion in Biological Sequence Design

**Hanqun Cao**[1]*, **Haosen Shi**[1]*, **Chenyu Wang**[2], **Sinno Jialin Pan**[1], **Pheng-Ann Heng**[1]†
[1]The Chinese University of Hong Kong [2]Massachusetts Institute of Technology

## Abstract

Diffusion models have emerged as powerful tools for biological sequence design, offering flexible conditional generation for engineering functional biomolecules. While reinforcement learning (RL)-based fine-tuning enables multi-objective optimization on limited data, existing methods face a critical trade-off: gradient-free approaches suffer from training instability in discrete spaces, whereas gradient-based methods incur prohibitive computational costs. This trade-off severely limits their practical applicability in biological design tasks. We propose GLID$^2$E, a light-weight gradient RL framework that achieves stable and efficient fine-tuning of discrete diffusion models. Our key insight is to constrain the exploration space through a clipped likelihood mechanism while employing reward shaping to align generation with design objectives. This combination mitigates the inherent instabilities in RL-guided diffusion while maintaining computational efficiency. We demonstrate GLID$^2$E's effectiveness on DNA and protein sequence design benchmarks, where it matches or exceeds the performance of gradient-based methods while requiring significantly lower computational resources. Our approach provides a practical solution for function-driven biological sequence optimization. The code is available at: `https://github.com/chq1155/GLID2E`.

## 1 Introduction

Designing biological sequences with desired functional properties is fundamental to protein engineering and synthetic biology [1, 2]. Recent diffusion [3, 4, 5, 6, 7] and flow-matching models [8, 9, 10, 11] have shown impressive capability in modeling sequence distributions. However, adapting these pretrained models for controllable, task-specific design remains challenging, particularly when limited experimental data inadequately captures sequence-function relationships and design objectives involve multiple competing criteria.

Two primary paradigms have emerged for functional sequence design: conditional sampling and fine-tuning. Conditional sampling methods [12, 13] guide generation online by steering the diffusion process toward desired properties. While conceptually straightforward, they incur additional inference costs and struggle to balance multiple objectives effectively. Fine-tuning methods [14, 15, 16, 17, 18] offer a complementary approach by embedding functional knowledge into model parameters. DRAKES [14], a representative gradient-based method, employs Gumbel-Softmax to enable gradient flow through discrete trajectories and uses KL regularization for distribution alignment. Although fine-tuned models generate functional sequences efficiently at inference without additional costs, gradient-based training poses significant challenges: backpropagation through entire generation trajectories requires storing multiple intermediate states, leading to substantial memory overhead and computational burden. Moreover, terminal-only rewards provide limited guidance, missing opportunities to leverage intermediate information for finer generation control.

---

*HC and HS contributed equally.
†Correspondence to: pheng@cse.cuhk.edu.hk

39th Conference on Neural Information Processing Systems (NeurIPS 2025).

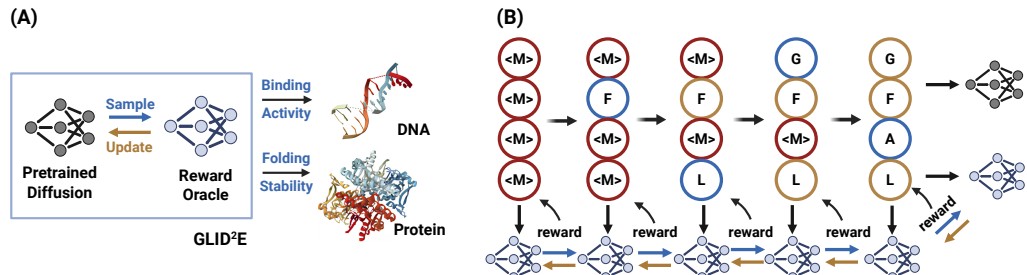

Figure 1: Overview of the GLID$^2$E framework. (A) GLID$^2$E employs a lightweight reinforcement learning approach to fine-tune pretrained discrete diffusion models for generating functional biological sequences, specifically regulatory DNA elements and thermostable proteins. (B) The framework incorporates two key innovations: a Clipped Likelihood Constraint that preserves sample rationality while allowing exploration, and a Reward Shaping mechanism that provides informative signals throughout the diffusion process, enabling stable and lightweight fine-tuning.

We propose GLID$^2$E (**G**radient **LI**ghtweight fine-tuning for **D**iscrete sequence **DE**sign), a reinforcement learning framework that reformulates fine-tuning as policy optimization. Unlike gradient-based methods that backpropagate through full trajectories, GLID$^2$E computes lightweight gradients only on policy parameters while treating the generation process as environment interactions. Our approach introduces two key mechanisms for stable and efficient optimization. First, we employ a *clipped likelihood constraint* that leverages the pretrained model's likelihood estimates to assess sample quality, avoiding explicit KL divergence computation while preventing unreasonable sequence exploration. This constraint enables effective high-reward region exploration while maintaining distributional validity. Second, our *reward shaping mechanism* provides informative signals at intermediate generation steps by evaluating partial sequences, guiding the policy toward promising regions earlier in the process. This contrasts with terminal-only reward methods, enabling more efficient learning and finer-grained control over generation trajectories.

We validate GLID$^2$E on DNA and protein sequence design benchmarks, demonstrating comparable or superior performance to state-of-the-art methods while achieving significantly faster training and inference with reduced computational requirements. Ablation studies confirm that both the clipped likelihood constraint and reward shaping contribute substantially to stability and performance. Our results suggest that GLID$^2$E provides a practical and scalable solution for function-driven biological sequence optimization.

## 2 Related Work

**Discrete Diffusion Models for Biological Sequences**  Diffusion models have been successfully extended from continuous to discrete domains, enabling powerful generative modeling for biological sequences [19, 20, 5, 6]. Recent advances have demonstrated strong performance in protein structure generation [21, 22], RNA design [23], and DNA sequence modeling [24].

Key technical challenges distinguish discrete diffusion from its continuous counterpart. First, the corruption process requires carefully designed categorical transition matrices rather than Gaussian noise [19, 20], though auto-regressive variants [25] can sidestep explicit transition design. Second, iterative categorical sampling incurs significant computational costs, particularly for long sequences. While accelerated sampling techniques [26] and consistency models [27] have improved efficiency, the trade-off between fidelity and computational cost remains when training data is limited—a common scenario in biological applications where experimental characterization is expensive.

**Conditional Generation and RL Fine-Tuning**  Adapting pretrained diffusion models for task-specific objectives follows two primary paradigms. *Training-free guidance* methods steer the sampling process online through classifier guidance [28, 29], Sequential Monte Carlo [13], or classifier-free guidance [30]. These approaches offer flexibility without retraining but incur additional inference costs and struggle to balance multiple competing objectives [31, 32, 33], limiting their effectiveness for complex biological design tasks.

*RL Fine-tuning* methods provide an alternative by embedding functional knowledge directly into model parameters [17, 18, 16]. DRAKES [14] employs gradient-based optimization with Gumbel-Softmax reparameterization and KL regularization, achieving strong performance in biological sequence design. However, backpropagating through entire generation trajectories requires substantial memory and computation. More broadly, reinforcement learning has emerged as a powerful framework for optimizing generative models with non-differentiable objectives [34, 35, 36, 33, 37], particularly in text generation where reward-driven fine-tuning improves fluency and alignment. Recent work applies policy gradient methods to molecular and protein design [14, 38, 39], but efficiency and stability challenges persist in high-dimensional discrete spaces.

**Our Approach**    GLID$^2$E bridges these paradigms by reformulating fine-tuning as lightweight policy optimization. Unlike gradient-based methods that backpropagate through generation trajectories, we compute gradients only on policy parameters while treating sampling as environment interactions. Our clipped likelihood constraint replaces expensive KL divergence computation with efficient likelihood-based filtering, while reward shaping addresses sparse reward signals without requiring intermediate gradient flow. This design achieves the parameter-embedded knowledge benefits of fine-tuning with computational efficiency approaching training-free methods, providing a practical solution for function-driven biological sequence optimization.

## 3  Preliminary

### 3.1  Problem Formulation

We consider the task of adapting a pretrained generative model to produce sequences with high functional value. Formally, let $\mathcal{X} \subseteq \{1, \ldots, N\}^n$ denote the discrete sequence space, where $N$ is the vocabulary size and $n$ is the sequence length. Our goal is to transform a pretrained diffusion model $p_{\text{prior}}(x)$ into an optimized policy $p_\theta(x)$ that assigns higher probability to sequences $x$ with high reward $r(x)$, where $r : \mathcal{X} \to \mathbb{R}$ is a reward function evaluating functional properties.

This setting is motivated by a fundamental tension in biological sequence design: pretrained diffusion models, trained on large-scale real-world data, generate sequences that are structurally plausible but not necessarily functionally optimized for specific tasks. Conversely, reward models can evaluate task-specific properties but may assign high scores to invalid or unrealistic sequences when used directly for optimization. Our objective is to leverage both the distributional knowledge of the pretrained model and the task-specific guidance from the reward function.

### 3.2  Discrete Diffusion Models

We briefly review discrete diffusion models based on continuous-time Markov chains (CTMCs) [5, 6]. The forward process gradually corrupts a sequence $x_0 \in \mathcal{X}$ into a fully masked sequence $x_T$ over time $t \in [0, T]$ via a time-dependent transition rate matrix $Q(t)$. The transition matrix is typically handcrafted to ensure that at $t = T$, the sequence consists entirely of special MASK tokens, representing maximum corruption.

The generative model learns to reverse this corruption process. A neural network parameterized by $\theta$ is trained to approximate the reverse-time transition rates $\bar{Q}^\theta(t)$, enabling ancestral sampling from the masked state $x_T$ back to realistic sequences $x_0$. The reverse process follows the time-reversed CTMC:

$$\frac{dx_{T-t}}{dt} = \bar{Q}^\theta(T - t)x_{T-t} \tag{1}$$

where the model learns to predict appropriate tokens to replace MASK symbols at each timestep.

Throughout this work, we assume the pretrained diffusion model operates on fixed-length sequences and is unconditional, generating samples $x \sim p_{\text{prior}}(x)$ without additional conditioning inputs. Our method fine-tunes this pretrained model to incorporate functional objectives while preserving its ability to generate valid sequences.

Table 1: Comparative analysis of biological sequence design methods, evaluating key features and computational requirements. GLID$^2$E uniquely combines light-weight policy optimization with preserved sequence naturalness, while achieving lower training complexity ($O(N{\cdot}B{\cdot}T{\cdot}L{\cdot}d)$ vs. $O(N{\cdot}B{\cdot}T{\cdot}L{\cdot}d^2)$ for comparable methods). Notation: $N$ = training iterations, $B$ = batch size, $T$ = diffusion steps, $L$ = sequence length, $d$ = model dimension, $P$ = number of particles.

| Method | Feature Comparison | | | Computational Cost Comparison | | |
|---|---|---|---|---|---|---|
| | Requires Gradients | Preserves Naturalness | Theoretical Guarantees | Training | Inference | Memory |
| CG | ✓ | ✗ | ✓ | $O(1)$ | $O(T{\cdot}L{\cdot}d^2)$ | $O(d^2)$ |
| SMC | ✗ | ✓ | ✗ | $O(1)$ | $O(P{\cdot}T{\cdot}L{\cdot}d)$ | $O(P{\cdot}L{\cdot}d + d^2)$ |
| TDS | ✓ | ✗ | ✗ | $O(1)$ | $O(P{\cdot}T{\cdot}L{\cdot}d^2)$ | $O(P{\cdot}L{\cdot}d + d^2)$ |
| CFG | ✓ | ✗ | ✗ | $O(N{\cdot}B{\cdot}T{\cdot}L{\cdot}d^2)$ | $O(T{\cdot}L{\cdot}d)$ | $O(d^2)$ |
| DRAKES | ✓ | ✓ | ✓ | $O(N{\cdot}B{\cdot}T{\cdot}L{\cdot}d^2)$ | $O(T{\cdot}L{\cdot}d)$ | $O(T{\cdot}L{\cdot}d + d^2)$ |
| GLID$^2$E | ✗ | ✓ | ✗ | $O(N{\cdot}B{\cdot}T{\cdot}L{\cdot}d)$ | $O(T{\cdot}L{\cdot}d)$ | $O(d^2 + B{\cdot}L)$ |

# 4 Method

We present GLID$^2$E (**G**radient **LI**ghtweight fine-tuning for **D**iscrete sequence **DE**sign), a reinforcement learning framework that efficiently adapts pretrained discrete diffusion models for functional sequence design. Unlike gradient-based methods that backpropagate through entire generation trajectories, GLID$^2$E treats the diffusion sampling process as an RL environment and optimizes only the policy parameters with lightweight gradients. Our framework addresses two fundamental challenges: (1) preventing policy collapse while maximizing rewards, and (2) overcoming sparse terminal rewards through intermediate guidance. We introduce three core components: a clipped likelihood constraint for rationality preservation (Section 4.1), reward shaping for informative intermediate signals (Section 4.2), and a PPO-based optimization that integrates these mechanisms (Section 4.3).

## 4.1 Clipped Likelihood Constraint

**The Standard KL-Regularized Objective.** Directly maximizing expected rewards in RL often leads to training instabilities and policy collapse [18, 40]. A standard approach employs KL divergence regularization to constrain the optimized policy $p_\theta$ near a reference policy $p_{\text{prior}}$:

$$\max_\theta \mathbb{E}_{x \sim p_\theta}\left[r(x)\right] - \beta\,\text{KL}(p_\theta \| p_{\text{prior}}), \tag{2}$$

where $\beta > 0$ controls the regularization strength. The optimal solution to Equation (2) takes the form:

$$p_{\theta^*}(x) \propto p_{\text{prior}}(x) \exp\left(\frac{r(x)}{\beta}\right). \tag{3}$$

This formulation presents a fundamental trade-off. Large $\beta$ yields conservative policies that closely follow the prior but achieve limited reward improvement—problematic when the prior distribution, trained on diverse task-agnostic data (e.g., entire proteomes or multi-species genomes), lacks task-specific inductive biases. Conversely, small $\beta$ permits aggressive exploration but risks generating invalid sequences when reward models exhibit pathologies, such as assigning artificially high scores to sequences with excessive hydrophobic regions while ignoring critical functional constraints.

**Rethinking the Constraint.** The optimal policy in Equation (3) uniformly mixes the prior and reward-induced distributions across all sequences. However, we argue that *rationality should be enforced as a hard constraint rather than softly blended with reward optimization*. Pretrained diffusion models, trained on extensive real-world datasets, inherently capture distributional validity and can effectively assess sequence plausibility through likelihood estimates. Meanwhile, over-reliance on the prior may unnecessarily restrict exploration of high-reward regions, particularly for task-specific objectives where the prior provides limited guidance.

This motivates reformulating the optimization as:

$$\begin{aligned}
\max_\theta \quad & \mathbb{E}_{x \sim p_\theta}[r(x)] - \alpha\,H(p_\theta) \\
\text{subject to} \quad & p_\theta(x) > 0 \text{ only if } p_{\text{prior}}(x) \geq c,
\end{aligned} \tag{4}$$

where $c > 0$ is a likelihood threshold defining the rationality boundary, and the entropy term $H(p_\theta) = -\mathbb{E}_{x \sim p_\theta}[\log p_\theta(x)]$ encourages policy diversity. This formulation restricts the policy's support to sequences deemed plausible by the pretrained model while allowing the reward function to dominate within this constrained space.

**Practical Implementation via Likelihood Clipping.** Solving Equation (4) directly is intractable. We propose a practical approximation by modifying the reward function to penalize sequences with low prior likelihood. We first generate a calibration set of $n_{\text{cal}}$ samples from the pretrained model and compute the empirical mean $\mu$ and standard deviation $\sigma$ of their log-likelihoods. The threshold is set as $c = \mu - k\sigma$, where $k \geq 0$ controls tolerance (we use $k = 1$ in experiments). This calibration leverages the pretrained model's confidence: samples within one standard deviation are considered plausible.

The modified reward function incorporates a likelihood-based penalty:

$$\tilde{r}(x) = r(x) + \beta \, \min\left( \frac{\log p_{\text{prior}}(x) - \mu}{\sigma} + k, 0 \right), \tag{5}$$

where the $\min(\cdot, 0)$ operator activates penalties only for sequences below the threshold (i.e., $\log p_{\text{prior}}(x) < \mu - k\sigma$), and $\beta > 0$ controls penalty strength. The standardization $(\log p_{\text{prior}}(x) - \mu)/\sigma$ ensures robustness across different models and tasks by normalizing likelihood scales.

Using $\tilde{r}(x)$ as the reward, we optimize the policy via standard RL without explicit KL regularization. This approach provides a computationally efficient alternative that avoids expensive KL divergence calculations while effectively constraining the policy to the high-likelihood region of the pretrained distribution.

## 4.2 Reward Shaping

**Motivation.** Standard RL formulations for diffusion models provide rewards only upon complete sequence generation, resulting in sparse signals that slow learning. To accelerate optimization and provide finer-grained guidance, we introduce reward shaping [41], a technique that assigns intermediate rewards while preserving the optimal policy.

**RL Formulation.** Following [14], we formulate diffusion sampling as a Markov Decision Process (MDP). The state space comprises partially denoised sequences $s_t = x_t$ at timestep $t \in \{0, \ldots, T\}$, where $x_0$ is fully masked and $x_T$ is the final sequence. At each step, the diffusion model performs a denoising action by sampling from $\pi_\theta(\cdot | x_{t-1}, t)$, transitioning from $s_{t-1}$ to $s_t$. The standard reward structure assigns zero rewards to all intermediate transitions ($r(s_t) = 0$ for $t < T$) and provides the final reward $r(s_T) = \tilde{r}(x_T)$ only at termination. The discount factor is $\gamma = 1.0$.

**Potential-Based Shaping.** We employ potential-based reward shaping [41], which modifies rewards via a potential function $\Phi : S \rightarrow \mathbb{R}$ as:

$$r'(s_{t-1}, s_t) = r(s_{t-1}, s_t) + \gamma \Phi(s_t) - \Phi(s_{t-1}). \tag{6}$$

This formulation guarantees that the cumulative return remains unchanged: $\sum_{t=1}^{T} r'(s_{t-1}, s_t) = r(s_T) + \Phi(s_T) - \Phi(s_0)$. By setting $\Phi(s_0) = \Phi(s_T) = 0$, the optimal policies under $r$ and $r'$ coincide.

**Handling Masked Tokens.** A key challenge is defining $\Phi(s_t)$ for intermediate states containing MASK tokens, which the reward model cannot evaluate directly since it was trained only on complete sequences. We address this by completing partial sequences through Monte Carlo sampling: for each intermediate state $s_t$ with $n_{\text{mask}}$ remaining masks, we sample $n_{\text{mc}}$ completions by independently replacing each MASK token according to the current policy $\pi_\theta(\cdot | x_t, t)$, with the MASK token probability set to zero. The potential function is defined as:

$$\Phi(s_t) = \begin{cases} \mathbb{E}_{x_t^{\text{comp}} \sim \pi_\theta^{\text{complete}}}[\tilde{r}(x_t^{\text{comp}})] & 1 \leq t < T \\ 0 & t = 0 \text{ or } t = T, \end{cases} \tag{7}$$

where $\pi_\theta^{\text{complete}}$ denotes the completion distribution. In practice, we approximate the expectation using $n_{\text{mc}}$ samples.

This design provides increasingly accurate reward estimates as denoising progresses: early states with many masks yield coarse estimates, while near-complete sequences provide precise signals. The

shaped rewards guide the policy toward promising regions throughout the trajectory, accelerating learning compared to terminal-only feedback.

## 4.3 PPO-Based Policy Optimization

We integrate the clipped likelihood constraint and reward shaping within a Proximal Policy Optimization (PPO) framework [42], a widely adopted policy gradient method known for stable training. PPO employs a clipped surrogate objective to prevent excessively large policy updates that could destabilize training.

**Mixed Reference Policy.** To maintain proximity to the pretrained diffusion model while allowing reward-driven exploration, we define a mixed reference policy that interpolates between the pretrained policy $\pi_{\text{prior}}$ and the current policy $\pi_\theta$:

$$\log \pi_{\text{ref}}(s_t|s_{t-1}) = \eta \log \pi_{\text{prior}}(s_t|s_{t-1}) + (1 - \eta) \log \pi_\theta(s_t|s_{t-1}), \tag{8}$$

where $\eta \in [0, 1]$ controls the mixture weight. This design creates a "soft anchor" to the pretrained distribution: higher $\eta$ enforces stronger adherence to the prior, while lower $\eta$ grants more exploration freedom. During online sampling, we record logits from both policies to compute $\pi_{\text{ref}}$ efficiently.

**Clipped Surrogate Objective.** The PPO objective clips importance ratios to bound policy updates:

$$L^{\text{CLIP}}(\theta) = \mathbb{E}_{t,s_t} \left[ \min \left( \rho_t \hat{A}_t, \text{clip}(\rho_t, 1 - \epsilon, 1 + \epsilon) \hat{A}_t \right) \right], \tag{9}$$

where $\rho_t = \pi_\theta(s_t|s_{t-1})/\pi_{\text{ref}}(s_t|s_{t-1})$ is the importance weight, $\hat{A}_t$ is the advantage estimate computed via Generalized Advantage Estimation (GAE) [43], and $\epsilon > 0$ is the clipping threshold (set to $0.5$ in our experiments, corresponding to a clipping ratio of $1.5$). The clipping mechanism prevents large policy shifts while avoiding expensive KL divergence computations.

Combined with an entropy bonus to encourage exploration, the final objective becomes:

$$L(\theta) = L^{\text{CLIP}}(\theta) + \lambda H(\pi_\theta), \tag{10}$$

where $\lambda > 0$ controls the entropy coefficient. Together with the clipped likelihood constraint (Equation 5) and reward shaping (Equation 6), this framework achieves efficient and stable fine-tuning of discrete diffusion models. The complete algorithm is provided in Appendix A.4, with hyperparameter details in Appendix A.1.

## 5 Experiments

We evaluate GLID$^2$E on two biological sequence design benchmarks: regulatory DNA enhancer design and thermostable protein design. Our experiments demonstrate that GLID$^2$E achieves state-of-the-art or competitive performance while maintaining computational efficiency. We conduct comprehensive ablation studies to validate the contribution of each component and analyze the impact of key hyperparameters.

### 5.1 Experimental Setup

**Datasets and Models**  Following DRAKES [14], we use established benchmarks for both tasks.

**DNA Design.** We use a comprehensive enhancer dataset containing approximately 700,000 DNA sequences of 200 base pairs [44], characterized for activity in human cells via massively parallel reporter assays. The discrete diffusion model follows [45], while the reward oracle adopts the Enformer architecture [46] trained to predict activity in the HepG2 cell line.

**Protein Design.** The discrete diffusion model is pretrained on 19,700 high-resolution single-chain structures from PDB, following ProteinMPNN [2]. The reward oracle is trained on the Megascale dataset (1.8M sequences across 983 designed domains) to predict thermodynamic stability measured by Gibbs free energy change ($\Delta\Delta G$). Both models use the ProteinMPNN architecture.

**Evaluation Metrics** We use metrics for both functional performance and naturalness preservation.

**DNA Design.** Functionality is measured by *Pred-Activity* (predicted enhancer activity) and *ATAC-Acc* (chromatin accessibility [47, 48]). Naturalness is quantified via *3-mer Correlation* (similarity to natural k-mer distributions) and *Log-Likelihood* under the pretrained model [44].

**Protein Design.** Stability is assessed by *Pred-ddG* (predicted $\Delta\Delta G$) and *%(ddG>0)* (percentage of stabilizing sequences). Structural naturalness is evaluated using *scRMSD* (self-consistency RMSD between ESMFold [49] predictions from sequence and back-translation). We define *Success Rate* as the percentage of sequences satisfying both ddG>0 and scRMSD<2.

**Baselines** We compare against the pretrained diffusion model, conditional sampling methods (Classifier Guidance [29], SMC [13], TDS [50], and Classifier-Free Guidance [30]), the MCTS-based tree method PepTune [51], and the gradient-based fine-tuning method DRAKES [14].

**Implementation Details** All experiments are conducted on a single NVIDIA A40 GPU with 20GB memory. We use multiple GPUs for parallel runs across different random seeds. Hyperparameters are detailed in Appendix A.1.

## 5.2 DNA Sequence Design

Table 2: General performance for DNA sequence design models. State-of-the-art performance is **bold**, and the second-highest performance is underlined. KL, M1, and M2 denote KL regularization, reward shaping, and likelihood penalty, respectively.

| Method | Pred-Activity (median)↑ | ATAC-Acc↑(%) | 3-mer Corr↑ | Log-Lik (median)↑ |
|---|---|---|---|---|
| Pretrained | 0.17(0.04) | 1.5(0.2) | -0.061(0.034) | -261(0.6) |
| CG | 3.30(0.00) | 0.0(0.0) | -0.065(0.001) | -266(0.6) |
| SMC | 4.15(0.33) | 39.9(8.7) | 0.840(0.045) | -259(2.5) |
| TDS | 4.64(0.21) | 45.3(16.4) | 0.848(0.008) | -257(1.5) |
| CFG | 5.04(0.06) | 92.1(0.9) | 0.746(0.001) | -265(0.6) |
| DRAKES | 5.61(0.07) | 92.5(0.6) | **0.887(0.002)** | -264(0.6) |
| GLID$^2$E | **7.29(0.162)** | **98.4(0.67)** | 0.49(0.074) | -240.933(3.7) |
| GLID$^2$E  w/o M1 | 2.57(0.60) | 0.63(0.3) | 0.473(0.078) | **-239.12(10.07)** |
| GLID$^2$E  w/o M2 | 6.62(0.42) | 67.3(39.4) | 0.458(0.009) | -244.65(21.5) |

**Main Results** Table 2 presents the DNA design results. GLID$^2$E achieves the highest median Pred-Activity (7.29), substantially outperforming DRAKES by 30% and demonstrating superior functional optimization. Our method also attains the best ATAC-Acc (98.4%), indicating that generated sequences exhibit high chromatin accessibility, a key indicator of functional enhancer activity in biological contexts.

Regarding naturalness metrics, GLID$^2$E achieves competitive Log-Likelihood (-240.9), second only to the ablation variant without M1, confirming that our clipped likelihood constraint effectively preserves distributional validity. However, GLID$^2$E exhibits lower 3-mer correlation (0.49) compared to DRAKES (0.887) and conditional sampling methods. This discrepancy reveals an important finding: the pretrained model's learned distribution diverges from conventional k-mer statistics of natural enhancers. Rather than indicating lower quality, this suggests GLID$^2$E discovers functionally equivalent but compositionally distinct sequence variants that satisfy the model's likelihood constraints while achieving superior predicted activity. This exploration beyond traditional motif patterns potentially expands the design space for functional enhancers.

GLID$^2$E exhibits lower variance in activity metrics (Pred-Activity std: 0.162) compared to naturalness metrics (Log-Lik std: 3.7), suggesting convergence toward reward-optimized regions while maintaining diversity in sequence-level characteristics.

**Ablation Study** The ablation experiments validate our design choices. Removing reward shaping (w/o M1) causes dramatic performance drops: Pred-Activity decreases to 2.57 (65% reduction) and ATAC-Acc collapses to 0.63%, demonstrating that intermediate reward signals are critical for guiding

the policy toward functional regions during early generation steps. Notably, Log-Likelihood remains comparable (-239.1), indicating that the model still generates valid sequences but fails to optimize for functionality without shaped rewards.

Removing the likelihood constraint (w/o M2) maintains reasonable activity (Pred-Activity: 6.62) but reduces ATAC-Acc to 67.3% and degrades Log-Likelihood to -244.7, confirming that the clipped likelihood mechanism preserves both distributional validity and biologically relevant sequence characteristics. This validates our hypothesis that the pretrained model's likelihood estimates capture essential naturalness constraints beyond simple k-mer statistics.

### 5.3 Protein Sequence Design

Table 3: General performance for protein sequence design models. State-of-the-art performance is **bold**, and the second-highest performance is underlined. KL, M1, and M2 denote KL regularization, reward shaping, and likelihood penalty, respectively.

| Method | Pred-ddG (median) ↑ | %(ddG > 0) (%) ↑ | scRMSD (median) ↓ | %(scRMSD < 2) (%) ↑ | Success Rate (%) ↑ |
|---|---|---|---|---|---|
| Pretrained | -0.544(0.037) | 36.6(1.0) | 0.849(0.013) | 90.9(0.6) | 34.4(0.5) |
| CG | -0.561(0.045) | 36.9(1.1) | 0.839(0.012) | 90.9(0.6) | 34.7(0.9) |
| SMC | 0.659(0.044) | 68.5(3.1) | 0.841(0.006) | 93.8(0.4) | 63.6(4.0) |
| TDS | 0.674(0.086) | 68.2(2.4) | **0.834(0.001)** | **94.4(1.2)** | 62.9(2.8) |
| CFG | -1.186(0.035) | 11.0(0.4) | 3.146(0.062) | 29.4(1.0) | 1.3(0.4) |
| DRAKES | **1.095(0.026)** | 86.4(0.2) | 0.918(0.006) | 91.8(0.5) | **78.6(0.7)** |
| PepTune | 0.432(0.003) | 94.0(2.0) | 1.041(0.057) | 87.3(4.0) | 70.1(1.2) |
| GLID$^2$E | 1.012(0.094) | **86.7(2.4)** | 0.961(0.047) | 89.2(1.2) | 76.7(1.2) |
| GLID$^2$E  w/o M1 | 0.843(0.099) | 75.3(2.5) | 0.950(0.074) | 85.0(3.0) | 62.0(3.4) |
| GLID$^2$E  w/o M2 | 0.893(0.170) | 80.6(3.9) | 0.970(0.049) | 85.8(5.1) | 67.5(4.4) |

**Main Results**    Table 3 shows protein design results. GLID$^2$E achieves competitive stability performance with Pred-ddG of 1.012 (second to DRAKES' 1.095) and the highest %(ddG>0) at 86.7%, demonstrating that RL-based fine-tuning can match gradient-based methods for thermodynamic optimization. Peptune exhibits a more concentrated ddG distribution, and its RMSD distribution shows similar performance to DRAKES and GLID$^2$E. The success rate of 76.7% approaches DRAKES' 78.6%, confirming the viability of our lightweight gradient approach. The

GLID$^2$E exhibits slightly higher scRMSD (0.961) compared to conditional sampling methods like TDS (0.834), reflecting broader exploration of sequence space. We note that scRMSD, while informative, provides a less direct naturalness measure than DNA's Log-Likelihood because ESMFold-predicted structural similarity may not fully capture sequence-level naturalness—a minor limitation of the evaluation framework. The higher variance across GLID$^2$E's metrics (e.g., Pred-ddG std: 0.094 vs. DRAKES' 0.026) further confirms this exploratory behavior, consistent with RL methods' tendency to sample more diverse regions of the design space.

**Ablation Study**    Removing reward shaping (w/o M1) reduces Pred-ddG to 0.843 and success rate to 62.0%, though performance still exceeds all conditional sampling methods. This confirms that iterative RL-based optimization accumulates more effective learning compared to one-shot conditional generation, and that shaped rewards effectively address sparse reward signals by providing informative gradients throughout the generation trajectory.

Removing the likelihood constraint (w/o M2) causes consistent degradation across all metrics (Pred-ddG: 0.893, success rate: 67.5%). Since the pretrained model was trained on natural, thermostable PDB structures, sequences closer to this distribution inherently achieve better stability and lower scRMSD. This validates that the clipped likelihood constraint serves dual purposes: preventing invalid sequence generation while implicitly guiding toward stable, natural structures. Unlike DRAKES' KL regularization, which uniformly constrains the entire distribution, our clipping mechanism permits controlled deviation (within one standard deviation), effectively filtering irrational outliers while allowing exploration of high-reward regions.

## 5.4 Hyperparameter Analysis

Table 4: Ablation study for clipping ratio in protein sequence design models. State-of-the-art performance is **bold**, and the second-highest performance is underlined.

| Clipping Ratio | Pred-ddG (median) ↑ | %(ddG > 0) (%) ↑ | scRMSD (median) ↓ | %(scRMSD < 2) (%) ↑ | Success Rate (%) ↑ |
|---|---|---|---|---|---|
| 0.5 | 0.402(0.147) | 65.6(0.5) | 1.022(0.083) | 87.0(1.3) | 53.9(1.3) |
| 1.0 | 0.834(0.281) | 75.4(11.2) | 0.988(0.061) | 81.7(5.1) | 60.5(1.8) |
| 1.5 | **1.012(0.094)** | **86.7(2.4)** | 0.961(0.047) | **89.2(1.2)** | **76.7(1.2)** |
| 2.0 | 0.932(0.144) | 76.4(2.7) | **0.902(0.041)** | 78.2(3.5) | 64.7(2.5) |

**Clipping Ratio** Table 4 shows the effect of the clipping ratio (recall that the likelihood threshold is $\mu - k\sigma$ where $k$ is the ratio). Stability metrics (Pred-ddG) initially increase then decrease: performance peaks at ratio 1.5, then declines at 2.0. This non-monotonic trend reveals a fundamental trade-off. Lower ratios (0.5-1.0) impose stricter likelihood constraints, limiting exploration of stable sequences beyond the pretrained distribution. Higher ratios (2.0) relax constraints excessively, allowing the policy to venture into regions where the pretrained model's likelihood estimates become less reliable.

Interestingly, as the clipping ratio increases, median scRMSD decreases (1.022→0.902) while %(scRMSD<2) also decreases (87.0%→78.2%), indicating emergence of a bimodal distribution: tighter likelihood constraints compress the policy toward the prior, yielding more sequences with excellent structural consistency, but also producing outliers with degraded structures. The high variance at ratio 1.0 (Pred-ddG std: 0.281, %(ddG>0) std: 11.2) reflects this transition point where the model struggles to balance reward optimization and distributional validity.

Table 5: Ablation study for mixture ratio in protein sequence design models. State-of-the-art performance is **bold**, and the second-highest performance is underlined.

| Mixture Ratio | Pred-ddG (median) ↑ | %(ddG > 0) (%) ↑ | scRMSD (median) ↓ | %(scRMSD < 2) (%) ↑ | Success Rate (%) ↑ |
|---|---|---|---|---|---|
| 0.01 | **1.012(0.094)** | **86.7(2.4)** | 0.961(0.047) | 89.2(1.2) | **76.7(1.2)** |
| 0.1 | 0.295(0.151) | 61.7(3.5) | 0.898(0.035) | 87.5(2.4) | 51.7(1.2) |
| 1.0 | -0.302(0.023) | 42.1(0.6) | **0.844(0.008)** | **93.4(1.2)** | 41.3(0.6) |

**Mixture Ratio** Table 5 examines the mixture ratio $\eta$ in the reference policy (Equation 8). Higher $\eta$ values enforce stronger adherence to the pretrained policy, resulting in progressively conservative behavior: at $\eta = 1.0$, performance degrades to Pred-ddG of -0.302 and success rate of 41.3%, barely improving over the pretrained baseline. This confirms that excessive regularization prevents effective reward optimization.

Conversely, low $\eta$ (0.01) permits aggressive exploration, achieving the best stability metrics (Pred-ddG: 1.012, success rate: 76.7%). The intermediate ratio ($\eta = 0.1$) exhibits substantially elevated variance across all metrics (e.g., Pred-ddG std: 0.151), similar to the clipping ratio analysis at 1.0. This suggests that moderate regularization creates ambiguity in the optimization landscape: the policy oscillates between prioritizing reward signals and adhering to the prior, leading to bimodal behavior that generates either high-reward or high-naturalness sequences without effectively balancing both objectives. These hyperparameter studies reveal that GLID$^2$E's performance is robust within reasonable ranges (clipping ratio: 1.5-2.0, mixture ratio: 0.01-0.1), with clear indicators (elevated variance) when hyperparameters approach suboptimal regimes.

## 6 Conclusion

We present GLID$^2$E, a reinforcement learning framework that adapts discrete diffusion models for functional biological sequence design through two key innovations: a clipped likelihood constraint that preserves distributional validity without expensive KL computation, and reward shaping that provides intermediate guidance throughout generation. These mechanisms enable stable and efficient

fine-tuning while maintaining parameter-level knowledge embedding. Experiments demonstrate that GLID$^2$E matches or exceeds state-of-the-art performance on DNA enhancer and protein design benchmarks. Our method achieves 30% activity improvement on DNA tasks and competitive stability (Pred-ddG: 1.012, success rate: 76.7%) on protein tasks, while requiring significantly lower computational resources. Ablation studies confirm both mechanisms contribute substantially to performance. GLID$^2$E's computational efficiency and modular design make it well-suited for multi-objective optimization, longer sequences, and data-scarce domains, providing a practical foundation for function-driven biological sequence design.

**Acknowledgements**    The work described in this paper was supported in part by the Research Grants Council of the Hong Kong Special Administrative Region, China, under Project T45-401/22-N; and in part by Guangdong-Hong Kong-Macao Joint Laboratory of Human-Machine Intelligence-Synergy Systems. Haosen Shi and Sinno J. Pan thank the support from the JC STEM Lab of Machine Learning and Symbolic Reasoning funded by The Hong Kong Jockey Club Charities Trust

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

# A  Appendix

## A.1  Implementation details

**Common Hyperparameters:**  In our implementation, we utilized several key hyperparameters that were common across experiments. These parameters control various aspects of our reinforcement learning approach, particularly focusing on reward handling and likelihood clipping mechanisms.

For reward processing, we employed parameters such as `likelihood penalty scale` to adjust the strength of likelihood constraint $\beta$ in Section 4.1 and `multiple_reward_sampling` to determine the number of multiple reward sampling for reward shaping in Section 4.2. For our clipped policy optimization, we use `clip_mix_factor` and `clip_threshold` to control the mixture, described in Section 4.3. The complete set of common hyperparameters and their values is presented in the table below:

Table 6: Common hyperparameter configuration utilized across experiments.

| Parameter | Setting |
|---|---|
| likelihood penalty scale | 0.1 |
| multiple_reward_sampling | 4 |
| clip_threshold | 1.5 |
| clip_mix_factor | 0.01 |

**Hyperparameters in DNA experiment:**  For our DNA experiments, we configured reinforcement learning parameters including GAE's $\lambda = 0.95$, discount factor ($\gamma = 0.99$), and learning rate ($1e-4$). We employed gradient norm clipping (1.0) and exponential moving average decay (0.999) to enhance training stability.

Table 7: Hyperparameter configuration utilized in our DNA experiments.

| Parameter | Setting |
|---|---|
| batch_size | 8 |
| decay | 0.999 |
| learning_rate | 1e-4 |
| $\lambda$ in GAE | 0.95 |
| $\gamma$ | 0.99 |
| gradient_norm_clip | 1.0 |
| gumbel_temperature | 1.0 |
| entropy_scale | 1e-3 |

**Hyperparameters in Protein experiment:**  Our protein experiments used 3 encoder and decoder layers with hidden dimension 128 and 30 neighbors for graph representation following [14]. We set the learning rate to 3e-5 with weight decay 1e-4 and used a diffusion process with 50 timesteps.

## A.2  Training time comparison

We compared the training times of DRAKES and GLID$^2$E, as shown in 9. The light-weight scheme enhances the algorithm's efficiency. We achieved consistent results with the training cost in Table 1.

## A.3  Sequence diversity analysis

We further tested the sequence diversity of different baselines based on entropy (Table 10). The results showed that all baselines exhibited a decline, with GLID$^2$E and CG achieving performance closest to that of the pretrained model.

Table 8: Hyperparameter configuration utilized in our protein experiments.

| Parameter | Setting |
|---|---|
| batch_size | 16 |
| hidden_dim | 128 |
| num_encoder_layers | 3 |
| num_decoder_layers | 3 |
| num_neighbors | 30 |
| dropout | 0.0 |
| backbone_noise | 0.1 |
| gradient_norm_clip | 1.0 |
| learning_rate | 3e-5 |
| weight_decay | 1e-4 |
| temperature | 0.1 |
| $\lambda$ in GAE | 0.95 |
| num_timesteps | 50 |
| gumbel_softmax_temperature | 0.5 |
| entropy_scale | 1e-3 |

Table 9: Training time per epoch for different methods.

| Method | **DRAKES** | **GLID$^2$E (Ours)** |
|---|---|---|
| Time | $23.28 \pm 0.14$ | $13.54 \pm 0.04$ |

## A.4   Detailed Training Algorithm of GLID$^2$E

**Algorithm 1** GLID$^2$E Training Algorithm based on PPO algorithm

---

1: **Input**: Policy network $p_\theta$, Value network $V_\phi$, Training epochs $K$, Advantage estimate $\hat{A}_t$, Clipping parameter $\epsilon$

2: **Initialization**: Policy network parameters $\theta$ and value network parameters $\phi$

3: **while** Termination condition is not met **do**

4:      Collect $N$ trajectories $\tau_i = (s_{i,t}, a_{i,t}, \log p(s_{i,t}))_{t=0}^T$, where $i = 1, \ldots, N$

5:      Calculate indicate states $x_{i,t}^b$

6:      Calculate reward $r_{i,t} = \Phi(x_{i,t+1}^b) - \Phi(x_{i,t}^b)$, where $t = 0, \ldots, T - 1$

7:      and $r_{i,T} = r(x_{i,T}) - \Phi(x_{i,T}^b) + \beta \min\left(\frac{\log p_{prior}(x_{i,T}) - \mu}{\sigma} + k, 0\right)$

8:      Compute advantage estimates $\hat{A}_{i,t}$ for each trajectory via GAE

9:      **for** $k = 1$ to $K$ **do**

10:         **for** Each mini-batch $B$ containing $M$ samples **do**

11:            Compute the policy loss $\mathcal{L}_{CLIP}(\theta)$

$$\mathcal{L}_{CLIP}(\theta) = -\hat{\mathbb{E}}_t \left[ \min\left( \frac{p_\theta(a_t|s_t)}{p_{\theta_{old}}(a_t|s_t)} \hat{A}_t, \text{clip}(\frac{p_\theta(a_t|s_t)}{p_{\theta_{old}}(a_t|s_t)}, 1 - \epsilon, 1 + \epsilon)\hat{A}_t \right) \right]$$

12:            Compute the value loss $\mathcal{L}_{VF}(\phi)$

$$\mathcal{L}_{VF}(\phi) = \hat{\mathbb{E}}_t \left[ \left( V_\phi(s_t) - \hat{V}_t \right)^2 \right]$$

where $\hat{V}_t$ is the estimated value

13:            Compute the entropy bonus $\mathcal{S}(\theta)$

$$\mathcal{S}(\theta) = \hat{\mathbb{E}}_t \left[ \mathcal{H} \left( \pi_\theta(\cdot|s_t) \right) \right]$$

where $\mathcal{H}$ is the entropy function

14:            Compute the total loss $\mathcal{L}(\theta, \phi)$

$$\mathcal{L}(\theta, \phi) = \mathcal{L}_{CLIP}(\theta) + \mathcal{L}_{VF}(\phi) - c_1 \mathcal{S}(\theta)$$

where $c_1$ is a hyperparameter

15:            Update policy network parameters $\theta$ and value network parameters $\phi$:

$$\theta \leftarrow \theta - \alpha_\theta \nabla_\theta \mathcal{L}(\theta, \phi)$$

$$\phi \leftarrow \phi - \alpha_\phi \nabla_\phi \mathcal{L}(\theta, \phi)$$

where $\alpha_\theta$ and $\alpha_\phi$ are learning rates

16:         **end for**

17:      **end for**

18: **end while**

---

Table 10: Diversity results based on sequence entropy.

| Method | Sequence Entropy ↑ |
|--------|--------------------|
| Pretrained | **34.7** |
| CG | 34.6 |
| SMC | 24.9 |
| TDS | 24.9 |
| CFG | 8.4 |
| DRAKES | 33.3 |
| GLID$^2$E | 34.6 |

## B   Discussions

### B.1   Limitations

While GLID$^2$E offers a lightweight yet competitive approach for DNA and protein sequence design, several limitations warrant acknowledgment. Algorithmically, reinforcement learning enables broader design space exploration but lacks the theoretical guarantees of methods like DRAKES, potentially yielding less stable fine-tuning and requiring more extensive hyperparameter tuning. Additionally, though presented as a comprehensive framework, GLID$^2$E requires validation across broader biological systems, including ligand-binding proteins, enzymes, antibodies, and RNA sequences, to fully demonstrate its robustness and versatility. Finally, our reliance on in silico validation cannot definitively establish real-world efficacy. Future wet lab experiments are essential to address this limitation and advance practical biological sequence design.

### B.2   Broader Impact

Our framework presents a versatile algorithm with broad applications in drug discovery systems, potentially accelerating therapeutic development and expanding discovery frontiers. However, this technology also poses misuse risks in designing harmful biological entities (proteins, RNA, DNA). Furthermore, AI-generated biological constructs raise important questions regarding intellectual property rights, patents, and ethical considerations that must be addressed as the field advances.

