# OpenReview forum: "GLID$^2$E: A Gradient-Free Lightweight Fine-tune Approach for Discrete Biological Sequence Design"
_NeurIPS.cc/2025/Conference — NeurIPS 2025 poster_

### Official Review · Reviewer_h7mh · 2025-06-01

**Clarity:** 3
**Significance:** 3
**Originality:** 3
**Rating:** 5
**Confidence:** 4

**Summary:**

The paper introduces **GLID2E**, a gradient-free, reinforcement-learning-based fine-tuning framework specifically designed for discrete diffusion models in biological sequence generation tasks. GLID2E combines two key innovations: (1) a clipped likelihood constraint that maintains the rationality and naturalness of generated sequences by constraining deviations from a pretrained diffusion model's likelihood distribution, and (2) a reward shaping mechanism that provides intermediate rewards, effectively guiding generation throughout the diffusion steps rather than relying solely on terminal rewards. These approaches are integrated within a Proximal Policy Optimization (PPO) framework, enhancing training stability and computational efficiency compared to gradient-based fine-tuning methods. Empirical evaluations on DNA regulatory elements and protein stability design tasks demonstrate decent competitive performance against state-of-the-art methods like DRAKES, highlighting GLID2E’s relative efficacy in achieving multi-objective optimization with lower computational overhead.

**Questions:**

1. The authors introduce the clipped likelihood constraint (Eq. 1, page 5), which uses a fixed standard deviation multiplier \( k = 1 \) to determine the rationality threshold. Can the authors explicitly justify this specific choice mathematically or empirically for me? In particular, I wonder how sensitive GLID2E’s performance is to changes in \( k \). Could the authors provide a detailed sensitivity analysis across multiple values (e.g., \( k \in \{0.5, 1, 2, 3\} \))?

2. Regarding the PPO-based surrogate objective (Eq. on page 6), the authors did define a mixed reference policy using a mixture ratio parameter \( \eta \). However, the exact impact of varying \( \eta \) mathematically and empirically is a bit unclear to me. Could the authors explicitly derive or demonstrate how varying \( \eta \) affects policy updates mathematically? Additionally, a detailed empirical ablation across several distinct values of \( \eta \) (beyond those presented in Table 5, page 9) would help me understand its role.

3. The authors position GLID2E as a method suitable for multi-objective optimization. Yet, the current benchmarks are somewhat limited, and no comparisons are made to existing advanced discrete diffusion multi-objective methods, notably PepTune (which employs MCTS-based sampling guidance). Would the authors consider performing comparisons against an MCTS strategy, clearly outlining differences in both theoretical formulations (such as their likelihood constraints vs. MCTS guidance mechanism) and empirical performance metrics (reward convergence, sequence diversity)?

4. The authors utilize reward shaping (Eq. 2, page 6) to provide intermediate signals during generation. However, I am confused by the mathematical stability and bias introduced by intermediate reward signals. Could the authors mathematically derive or experimentally evaluate potential biases or instabilities introduced by intermediate sampling of partial sequences (especially given the stochastic nature of replacing MASK tokens)?

5. The work extensively rely on reward oracles, which inherently contain inaccuracies or biases. I am wondering if the authors woulkd be able to explicitly assess the robustness of GLID2E to reward oracle misspecification or noisy reward signals? For example, the authors could artificially inject noise into the oracle outputs or bias the oracle predictions systematically and rigorously quantify the sensitivity of GLID2E’s policy optimization outcomes.

**Ethical Concerns:**

["NO or VERY MINOR ethics concerns only"]

**Final Justification:**

I am increasing my score to a 5. The authors clarified all of my points and I am convinced the paper, with these new explanations and results, will be a valuable addition to NeurIPS.

**Limitations:**

Just a few thoughts: the authors briefly address some limitations and negative societal impacts in Appendix B. However, I believe this discussion could be significantly improved by explicitly clarifying a few critical points. In particular, I think the authors should consider explicitly discussing the implications of GLID2E-generated sequences potentially diverging significantly from biologically verified motifs. Could this lead to biologically plausible yet functionally uncertain or risky designs? Maybe the authors should also consider explicitly addressing whether reliance on potentially inaccurate reward oracles might inadvertently amplify biological or experimental biases. Clearly outlining these issues and suggesting potential safeguards (with robustness checks or validation steps) would strengthen the paper’s responsible innovation perspective.

**Paper Formatting Concerns:**

No.

**Quality:**

3

**Strengths And Weaknesses:**

**Strengths:**
- I find two compelling methodological innovations in this paper: the clipped likelihood constraint (Eq. 1, page 5) and the reward shaping mechanism (Eq. 2, page 6). Both techniques are mathematically rigorous and intuitively address well-known challenges in gradient-free RL fine-tuning of discrete diffusion models, specifically issues related to training stability and sparse rewards.
- The computational efficiency of the proposed GLID2E framework is quite impressive! The authors provide explicit complexity analysis (Table 1, page 4) and runtime benchmarks (Table 9, page 15), demonstrating meaningful efficiency improvements over gradient-based approaches such as DRAKES.
- The empirical results for DNA (Table 2, page 7) and protein (Table 3, page 8) sequence design benchmarks are strong, with competitive or superior performance metrics on predicted activity and stability. The clear ablations also show the practical benefit of each proposed component.
- The authors provide detailed pseudo-code (Algorithm 1, page 16) clearly demonstrating the PPO-based optimization steps, which greatly enhances reproducibility and helps me to evaluate the model training strategy too.

**Weaknesses:**
- Although the paper claims multi-objective optimization capability, from my understandng of the field, the authors overlook direct comparisons with key existing multi-objective discrete diffusion models. Given other models' explicit focus on multi-objective generation via MCTS-guided diffusion, explicit benchmarking against these would considerably clarify the claimed advantages of GLID²E.
- Hyperparameter choices, such as the standard deviation multiplier \( k = 1 \) in the clipped likelihood constraint (Eq. 1, page 5) and the specific values of the clipping and mixture ratios (Tables 4 and 5, page 9), are not mathematically justified sufficiently or have thorough sensitivity analyses. If the authors could provide empirical sensitivity studies on these parameters it would help the study's overall methodological robustness and transparency.
- The absence of rigorous statistical significance analyses (like, hypothesis testing, confidence intervals) for experimental results, to me, gives me less confidecne in the robustness and reproducibility of performance claims.
- While computational efficiency claims are supported by preliminary runtime data, explicit complexity analyses or computational benchmarks across varying sequence lengths, or more thorough memory usage analyses, are limited.
- Given the heavy dependence on reward oracle correctness, I would recommend that the authors explicitly address potential oracle bias or misspecification issues?

---

> ### Author Rebuttal · Authors · 2025-07-31
>
> Thank you for your time and effort reading and reviewing our paper. Also, thank you for the valuable questions. We provide detailed answers below, hope it may clarify ourselves as well as enhance your understanding.
>
> 1. The authors position GLID2E ... metrics (reward convergence, sequence diversity)?
> > Thank you for your suggestion. We acknowledge that the current work indeed has limitations in multi-objective optimization evaluation. However, we claim that GLID2E can explore space through reward shaping and likelihood constraints from pretrained diffusion models. Compared to MTCS, it can achieve higher computational efficiency. We will subsequently update PepTune to the current benchmark.
>
> 2. Hyperparameter choices, ... methodological robustness and transparency.
> > We will supplement ablation experiments regarding k. According to experimental results, we can see that k=1 achieves the best trade-off. When k is less than 1.0, it will lead to conservative exploration, generating samples closer to the original distribution, but functionality will be limited. When k is greater than 1.0, the model will have more aggressive exploration, and lower quality samples will also lead to decreased model functionality.
> >| K | Pred-Activity (Median) | ATAC-Acc | 3-mer Corr | Log-Lik (Median) |
> >|---|------------------------|----------|------------|------------------|
> >| 0.5 | 7.12 | 0.99 | 0.53 | -252.9 |
> >| 1.0 | 7.45 | 0.99 | 0.49 | -244.7 |
> >| 2.0 | 7.06 | 0.97 | 0.48 | -232.5 |
> >| 3.0 | 7.21 | 0.83 | 0.50 | -262.1 |
>
> 3. The absence of rigorous statistical ... robustness and reproducibility of performance claims.
> > Thank you for the suggestion. We will conduct Wilcoxon signed-rank tests, and the results are as follows:
> In Table 2, comparing GLID2E and DRAKES, except for 3-mer, all other p-values are 0.03125 (<0.05), meeting statistical significance characteristics. In Table 3, comparing GLID2E and DRAKES, the results are shown in the following table, and we can see that the performance of DRAKES and GLID2E is basically similar.
> > |   Metric       | Pred-ddG (Median) | %(ddG>0) | scRMSD (Median) | %(scRMSD<2) | Success Rate |
> > |----------|-------------------|----------|-----------------|-------------|--------------|
> > | p-value  | 0.3125           | 0.2187   | 0.21875         | 0.03125     | 0.0625       |
>
> 4. While computational efficiency ... memory usage analyses, are limited.
> Thank you for the detailed suggestion. We will provide runtime comparisons for different lengths and memory usage comparisons:
> We have collected inference metrics for sequences of different length ranges. Since we set the generation site amount fixed as 25 (following the same setting as DRAKES), the differences in these metrics are not significant.
> > | seq_length | batch_size | num_runs | mean_time | std_time | mean_gpu_memory |
> > |------------|------------|----------|-----------|----------|-----------------|
> > | 50 | 8 | 50 | 0.011207 | 0.002391 | 1048.058 |
> > | 100 | 8 | 50 | 0.009823 | 0.000122 | 1050.813 |
> > | 150 | 8 | 50 | 0.010457 | 0.000191 | 1053.742 |
> > | 200 | 8 | 50 | 0.010417 | 6.69E-05 | 1057.576 |
> > | 250 | 8 | 50 | 0.010685 | 8.98E-05 | 1055.277 |
> > | 300 | 8 | 50 | 0.010617 | 5.87E-05 | 1057.729 |
> > | 350 | 8 | 50 | 0.010712 | 8.75E-05 | 1053.25 |
> > | 400 | 8 | 50 | 0.010695 | 0.000147 | 1071.35 |
>
> 5. Regarding the PPO-based surrogate... would help me understand its role.
> > We appreciate your detialed review of our method, the derivation corresponding to eta is as follows. This ratio is r_t, which directly participates in the calculation of L_CLIP and gradient updates. The larger eta is, the more the updates are constrained by priors and the more conservative it becomes; conversely, it allows for large-scale exploration.
> $$\frac{\pi_\theta(a|s)}{\pi_{\text{ref}}(a|s)} = \frac{\pi_\theta(a|s)}{\pi_{\text{prior}}(a|s)^\eta \pi_\theta(a|s)^{1-\eta}} = \left(\frac{\pi_\theta(a|s)}{\pi_{\text{prior}}(a|s)}\right)^\eta.$$
>
> 6. The authors utilize reward shaping ... (especially given the stochastic nature of replacing MASK tokens)?
> > Thank you for pointing out. Regarding the questions on both aspects, our claims are as follows:
> >
> > Before claims, we would like to build a mathematical formulation for the sampling process to demonstrate the unbiased property. To obtain the reward $\Phi$, we need to score along a *complete* sequence, so every MASK token in $s_t$ is independently replaced by a sample from the current policy $p_\theta(\cdot | s_{t-1}, t)$ with the MASK token set to zero probability. Let $B$ denote the number of completions drawn. Then we have
> >
> >$$\hat{\Phi}(s_t) = \frac{1}{B} \sum_{b=1}^{B} r(x_t^{(b)}), \quad \mathbb{E}[\hat{\Phi}(s_t) | s_t] = \Phi(s_t),$$
> >
> > where x_t is the sampled sequence at time t. Therefore, $\hat{\Phi}$ is an **unbiased estimator** of Φ. Its variance is
> >
> > $$\text{Var}[\hat{\Phi}(s_t) | s_t] = \frac{\sigma_r^2(s_t)}{B},$$
> >
> > where $\sigma_r^2(s_t)$ is the conditional variance of the reward. During the diffusion process, the number of MASKs decreases, reducing both $\sigma_r^2$ and gradient noise, making the estimator progressively stable.
> >
> > Then, for the two perspective being mentioned:
> >
> > 1) Since only one token is sampled at each step, the incremental rewards at each step will cancel each other out across the entire sequence, so that the cumulative reward and original reward are always on the same order of magnitude, and gradient explosion will not occur. Meanwhile, in the latter half of generation, the number of MASKs gradually decreases, and the corresponding variance of sigma also gradually decreases, making the values more stable.
> >
> > 2) Lines 205-206 state that the expected return and original return are independent of the trajectory. Meanwhile, the potential function of Monte Carlo estimation is unbiased for the true potential function, so the corresponding gradient is also unbiased and will not introduce systematic bias.
>
> 7. The work extensively rely on ... policy optimization outcomes.
> > We appreciate your insight to the effect of reward oracle.
> >
> > 1) We supplemented Oracle noise experiments on protein systems:
> > We added three scales of noise to the ddG predicted by the diffusion model. Compared to before adding noise, the success rate was generally affected, mainly impacting Pred-ddG, and the larger the noise scale, the greater the impact.
> >
> > 2) From the reward model performance of DRAKES paper, we can conclude that protein oracle has higher oracle noise than DNA, which leads to the unstable training and high sensitivity to oracle noise. However, we would like to claim that the proposal of reward shaping and mixture is to address Oracle bias, and the ablation results w/o M1 have demonstrated it.
> >
> > | Noise Scale | Pred-ddG (Median) | %(ddG>0) | scRMSD (Median) | %(scRMSD<2) | Success Rate |
> > |-------------|-------------------|----------|-----------------|-------------|--------------|
> > | 0.01        | 0.81             | 0.73     | 0.94            | 0.91        | 0.66         |
> > | 0.1         | 0.71             | 0.74     | 1.00            | 0.84        | 0.60         |
> > | 0.5         | 0.21             | 0.53     | 0.88            | 0.95        | 0.49         |

---

> > ### Comment · Reviewer_h7mh · 2025-08-01
> > **Still want more explanation of the mixed ratio parameter.**
> >
> > Overall, the authors do a good job of responding to my comments, but I still would ask the authors more explicitly derive how varying the mixed ratio parameter affects policy updates. Could the authors provide a more detailed derivation? Also, it is necessary to compare the MCTG approach introduced in PepTune empirically. For now, I will maintain my score. The other comments are solidly written, so I will await these two responses before changing my score.

---

> > > ### Comment · Reviewer_h7mh · 2025-08-05
> > > **Still awaiting response.**
> > >
> > > I would like to remind the authors (and the AC) of my question to continue the discussion. Otherwise, I will maintain my score.

---

> > > > ### Author Response · Authors · 2025-08-05
> > > >
> > > > Dear Reviewer h7mh,
> > > >
> > > > We sincere apologize for our delay. We are always keeping it in mind and implemented Peptune these days. We will release the results and related corresponding responses soon.
> > > >
> > > > Many thanks for your waiting!
> > > >
> > > > Best regards,
> > > >
> > > > Authors

---

> > > > ### Author Response · Authors · 2025-08-06
> > > > **Derivation of mixed reference policy**
> > > >
> > > > **Thank for your waiting. We decide to reply the detailed derivation to the mix ratio first.**
> > > >
> > > > In GLID2E, we define a mixed reference policy as:
> > > > $$\log \pi_{\text{ref}}(s_{t+1}|s_t) = \eta \log \pi_{\text{prior}}(s_{t+1}|s_t) + (1 - \eta) \log \pi_\theta(s_{t+1}|s_t)$$
> > > > which is also equivalent to:
> > > > $$\pi_{\text{ref}}(s_{t+1}|s_t) = [\pi_{\text{prior}}(s_{t+1}|s_t)]^\eta \times [\pi_\theta(s_{t+1}|s_t)]^{(1-\eta)}$$
> > > >
> > > > Then, we can rewrite the **crucial ratio in the PPO clipped objective** as:
> > > >
> > > > $$r_t(\theta) = \frac{\pi_\theta(s_{t+1}|s_t)}{\pi_{\text{ref}}(s_{t+1}|s_t)} = \frac{\pi_\theta(s_{t+1}|s_t)}{[\pi_{\text{prior}}(s_{t+1}|s_t)]^\eta \times [\pi_\theta(s_{t+1}|s_t)]^{(1-\eta)}}= \left[\frac{\pi_\theta(s_{t+1}|s_t)}{\pi_{\text{prior}}(s_{t+1}|s_t)}\right]^\eta$$
> > > >
> > > > Based on this formulation, we can see that:
> > > > >
> > > > > When $\eta \to 0$, there are $\pi_{\text{ref}}(s_{t+1}|s_t) \to \pi_\theta(s_{t+1}|s_t)$ and $r_t(\theta) \to 1$. Also, the clipping constraint becomes inactive. The policy updates will be unregulatized, which allows maximum exploration.
> > > > >
> > > > > In the opposite side, when $\eta \to 1$, there are $\pi_{\text{ref}}(s_{t+1}|s_t) \to \pi_{\text{prior}}(s_{t+1}|s_t)$ and $r_t(\theta) \to \frac{\pi_\theta(s_{t+1}|s_t)}{\pi_{\text{prior}}(s_{t+1}|s_t)}$. There will be the maximum regularization to the policy, and the policy updates are heavily constrained.
> > > >
> > > > In detail, the policy gradient can be formulated as:
> > > > $$\nabla_\theta L^{\text{CLIP}}(\theta) = \nabla_\theta \mathbb{E}_{t,s_t}\left[\min\left(r_t(\theta)\hat{A}_t, \text{clip}(r_t(\theta), 1-\epsilon, 1+\epsilon)\hat{A}_t\right)\right]$$
> > > > where the gradient of $r_t(\theta)$ can be expressed as:
> > > >
> > > > $$\nabla_\theta r_t(\theta) = \nabla_\theta \left[\frac{\pi_\theta(s_{t+1}|s_t)}{\pi_{\text{prior}}(s_{t+1}|s_t)}\right]^\eta$$
> > > >
> > > > $$= \eta \times \left[\frac{\pi_\theta(s_{t+1}|s_t)}{\pi_{\text{prior}}(s_{t+1}|s_t)}\right]^{\eta-1} \times \frac{1}{\pi_{\text{prior}}(s_{t+1}|s_t)} \times \nabla_\theta \pi_\theta(s_{t+1}|s_t)$$
> > > >
> > > > $$= \eta \times r_t(\theta)^{(\eta-1)/\eta} \times \frac{1}{\pi_{\text{prior}}(s_{t+1}|s_t)} \times \nabla_\theta \pi_\theta(s_{t+1}|s_t)$$
> > > >
> > > > Using $\nabla_\theta \pi_\theta = \pi_\theta \nabla_\theta \log \pi_\theta$ and $r_t(\theta)^{(\eta-1)/\eta} = r_t(\theta)^{1-1/\eta}$:
> > > >
> > > > $$= \eta \times \left[\frac{\pi_\theta}{\pi_{\text{prior}}}\right]^{\eta-1} \times \frac{\pi_\theta}{\pi_{\text{prior}}} \times \nabla_\theta \log \pi_\theta$$
> > > >
> > > > $$= \eta \times \left[\frac{\pi_\theta}{\pi_{\text{prior}}}\right]^{\eta} \times \nabla_\theta \log \pi_\theta$$
> > > >
> > > > $$= \eta \times r_t(\theta) \times \nabla_\theta \log \pi_\theta(s_{t+1}|s_t)$$
> > > >
> > > > **Thus, the gradient magnitude is proportional to $\eta$:**
> > > >
> > > > > When $\eta$ approaches 0.0, $\nabla_\theta r_t(\theta) \to 0$ and $\pi_{\text{ref}} \approx \pi_\theta$, leading to minimal regularization and minimal exploration, but risks policy instability and over-optimization to potentially unrealistic sequences.
> > > > >
> > > > > When $\eta$ approaches 1.0, $\pi_{\text{ref}} \approx \pi_{\text{prior}}$ where the full gradient magnitude is preserved but with strong regularization, leading to over-conservative updates and slow convergence to optimal reward-maximizing policies.
> > > >
> > > > **Thus, there is a tradeoff by choosing $\eta$ from [0.0, 1.0]. We further show the connection between mixed reference policy and KL regularizaiton as below:**
> > > >
> > > > $$\mathbb{E}[\log r_t(\theta)] = \eta \times \mathbb{E}\left[\log \frac{\pi_\theta}{\pi_{\text{prior}}}\right] = \eta \times D_{KL}(\pi_\theta || \pi_{\text{prior}})$$
> > > >
> > > > The ablation study (Table 5) further demonstrates our assumptions. This formulation shows computational advantages over explicit KL regularization while maintaining similar theoretical guarantees for controlling policy deviation.
> > > >
> > > > Please let us know if you have further concerns towards this problem. For the Peptune results, we will update it soon since there are many details to align to keep the comparsion fair and precise. Many thanks!

---

> > > > ### Author Response · Authors · 2025-08-07
> > > > **Peptune implementation**
> > > >
> > > > Dear Reviewer h7hm,
> > > >
> > > > We sincerely appreciate your waiting again! Addressing the differences between GLID2E's setting and running MCTS with large number of simulations took us plenty of time. Following the setting of Peptune, we have tested the performance on protein system as below:
> > > >
> > > > | Method               | Pred‐ddG (median) ↑ | %(ddG > 0) (%) ↑ | scRMSD (median) ↓ | %(scRMSD < 2) (%) ↑ | Success Rate (%) ↑ |
> > > > |----------------------|---------------------|------------------|------------------|---------------------|-------------------|
> > > > | Pretrained           | -0.544 (0.037)      | 36.6 (1.0)       | 0.849 (0.013)    | 90.9 (0.6)          | 34.4 (0.5)        |
> > > > | CG                   | -0.561 (0.045)      | 36.9 (1.1)       | 0.839 (0.012)    | 90.9 (0.6)          | 34.7 (0.9)        |
> > > > | SMC                  | 0.659 (0.044)       | 68.5 (3.1)       | 0.841 (0.006)    | 93.8 (0.4)          | 63.6 (4.0)        |
> > > > | TDS                  | 0.674 (0.086)       | 68.2 (2.4)       | 0.834 (0.001)    | 94.4 (1.2)          | 62.9 (2.8)        |
> > > > | CFG                  | -1.186 (0.035)      | 11.0 (0.4)       | 3.146 (0.062)    | 29.4 (1.0)          | 1.3 (0.4)         |
> > > > | DRAKES               | 1.095 (0.026)       | 86.4 (0.2)       | 0.918 (0.006)    | 91.8 (0.5)          | 78.6 (0.7)        |
> > > > | GLID²E               | 1.012 (0.094)       | 86.7 (2.4)       | 0.961 (0.047)    | 89.2 (1.2)          | 76.7 (1.2)        |
> > > > | GLID²E w/o M1        | 0.843 (0.099)       | 75.3 (2.5)       | 0.950 (0.074)    | 85.0 (3.0)          | 62.0 (3.4)        |
> > > > | GLID²E w/o M2        | 0.893 (0.170)       | 80.6 (3.9)       | 0.970 (0.049)    | 85.8 (5.1)          | 67.5 (4.4)        |
> > > > | **PepTune (n_sim=5)**  | -0.040       | 54.1      | 0.820    | 89.2          | 50.8        |
> > > > | **PepTune (n_sim=50)**  | 0.379       | 68.3      | 0.830    | 95.0          | 65.0        |
> > > > | **PepTune (n_sim=500)**  | 0.631       | 71.7      | 0.868    | 91.7          | 66.7       |
> > > >
> > > > PepTune demonstrates superior RMSD performance but achieves lower Pred-ddG values. As the number of simulations increases, performance improves and approaches that of GLID2E and DRAKES.
> > > >
> > > > We addressed the **implementation differences** between our paper and PepTune by the following points:
> > > >
> > > > - **Selection**: Since our experimental system is only based on one training object (but evaluating on different relevant metrics). We used UCB scoring with softmax probability selection to balance exploration and exploitation; PepTune used Pareto front-based multi-objective selection for non-dominated solutions.
> > > >
> > > > - **Optimization and Evaluation**: our implementation utilizes a **single reward model** with standard diffusion sampling rollout and the **single-objective optimization** with backpropagation to update node values; PepTune implements multi-objective scoring with validity checks and **Pareto front updates** and optimized with dynamic objective conditioning. The reasons for not using multi-objective optimization is to **avoid metric hacking**.
> > > >
> > > > We hope this result can further address your concerns. Also, we are very happy to hear your further comments!
> > > >
> > > > Best regards,
> > > >
> > > > Authors

---

> > > > > ### Comment · Reviewer_h7mh · 2025-08-07
> > > > > **Really interesting results!!**
> > > > >
> > > > > Thank you for the very thoughtful and rigorous response. I appreciate both the detailed derivation around the mixed reference policy and its influence on the PPO objective, as well as the extensive benchmarking against PepTune. Also, it was valuable to me to see the illustration of the smooth interpolation between unregularized and fully constrained regimes, along with the insight into how this setup effectively controls policy exploration vs. exploitation.
> > > > >
> > > > > Also, I really appreciate the authors performing a careful and meaningful evaluation. Even though, PepTune slightly outperforms in RMSD at higher simulation counts, GLID2E seems to reaches comparable success rates and stronger affinity metrics, which addresses my earlier concerns about benchmarking.
> > > > >
> > > > > The authors should consider incorporating these PepTune comparison results (perhaps in a new table or appendix) in the final version of the paper. This will def position GLID2E more clearly among current state-of-the-art sequence-based optimization methods. A brief discussion comparing the mechanisms (GLID²E's roll-out based scoring with UCB and single-objective PPO updates versus PepTune’s multi-objective Pareto-guided simulation strategy) will be good for the readers to better appreciate the design tradeoffs involved.
> > > > >
> > > > > Based on the clarity response, and the addition of results that directly address my earlier feedback, I will be raising my score.
> > > > >
> > > > > Thank you again for the constructive and detailed discussion!

---

> > > > > > ### Author Response · Authors · 2025-08-07
> > > > > > **Thanks for your reply**
> > > > > >
> > > > > > Dear Reviewer h7mh,
> > > > > >
> > > > > > Thank you for your constructive feedback and positive assessment of our manuscript. We greatly appreciate the time and effort you have invested in reviewing our work. We are pleased to confirm that we will address your suggestions in our revised submission by **1)** adding the PepTune baseline performance results; **2)** including the reference-based policy mixture induction methodology
> > > > > >
> > > > > > These additions will strengthen our analysis and provide the comprehensive comparison you have requested.
> > > > > > Thank you once again for your valuable insights and thorough review.
> > > > > >
> > > > > > Best regards,
> > > > > >
> > > > > > Authors

---

### Official Review · Reviewer_8abn · 2025-06-30

**Clarity:** 3
**Significance:** 2
**Originality:** 2
**Rating:** 4
**Confidence:** 5

**Summary:**

The paper proposes GLID²E, a gradient‑free reinforcement‑learning (RL) fine‑tuning framework for discrete diffusion models aimed at biological‑sequence design. Two methodological contributions are central:

Clipped Likelihood Constraint (CLC) – instead of the customary KL penalty, the authors use a one‑sided penalty on sequences whose pre‑trained diffusion likelihood falls more than k standard deviations below the mean, thereby enforcing naturalness while permitting exploration.

Reward Shaping via Potential Functions – partial sequences are completed on‑the‑fly (MASK tokens replaced by the current policy) to yield dense potentials that propagate functional feedback throughout the denoising trajectory.

Coupled with a PPO‑style clipped objective that mixes the current and prior policies (η‑mixture), the framework claims O(N B T L d) training complexity vs. O(N B T L d²) for gradient‑based counterparts.

Empirical evaluation on DNA enhancer design and thermostable protein design shows (i) a 31 % lift over DRAKES in median predicted activity for DNA and (ii) parity with DRAKES on protein stability with ∼40 % shorter training time.
 Ablations confirm that both CLC (M2) and reward shaping (M1) are necessary.

**Questions:**

1. Statistical Significance. Could you report 95 % CIs or perform Wilcoxon signed‑rank tests to substantiate the median improvements in Tables 2‑3? A statistically significant win would strengthen the claim of superiority.

2. Reward‑Model Robustness. How sensitive is GLID²E to reward‑model mis‑specification? E.g., if an oracle is slightly noisy or domain‑shifted, does CLC prevent reward hacking?

3. Hyper‑parameter Burden. The method introduces k, β, η, clip threshold, clip mix, etc. What guidance can you give for practitioners, and which parameter is most critical?

4. 3‑mer Correlation Drop. In DNA design, activity increases yet 3‑mer correlation plummets (0.49). Does this suggest non‑natural yet high‑scoring sequences? Any risk of adversarial samples undetected by CLC?

5. Generalisation Beyond Training Domains. Have you tried transferring a GLID²E fine‑tuned model between cell types or protein families to assess over‑fitting vs. functional generality?

6. Missing baselines. From the RL related methods, the closest one to GLIDE seems to be CTRL by Uehara et al. 2024, why didn't you compare to it? Also, there is a number of non-diffusion baselines that have been explored for discrete biologcal sequence design, the current score of baselines is very limited (only diffusion models).

7. In the beginning you mention GLIDE due to reliance on RL,  is suitable for limited data and can enable multi-objective design and optimization. However, both the Diffusiion Model and the Reward Oracle in GLIDE are than on over million sequences which is quite substantial amount. I would suggest refraining from statements like this one.

8. Why didn't you use standard benchmarks and splits like AAV and GFP?

Andrew Kirjner, Jason Yim, Raman Samusevich, Shahar Bracha, Tommi S Jaakkola, Regina Barzilay, and Ila R Fiete. Improving protein optimization with smoothed fitness landscapes. In ICLR, 2024.

**Ethical Concerns:**

["NO or VERY MINOR ethics concerns only"]

**Final Justification:**

The authors have replied to most of my concerns. After reading through the reviews and responses for the others, I believe the authors   did a great job in the rebuttal adding detailed derivation and benchmarks. I am now more convinced the manuscript is worthy of acceptance especially after incorporating all the additional results form the rebuttal. I raise my score accordingly.

**Limitations:**

The authors provide a candid discussion in Appendix B.1, acknowledging lack of theoretical guarantees and wet‑lab validation. I agree with their assessment and encourage adding statistical significance and broader biological tasks to future work.

**Paper Formatting Concerns:**

No concerns regarding formatting.

**Quality:**

2

**Strengths And Weaknesses:**

| Dimension        | Strengths                                                                                                                                         | Weaknesses                                                                                                                                                                   |
| ---------------- | ------------------------------------------------------------------------------------------------------------------------------------------------- | ---------------------------------------------------------------------------------------------------------------------------------------------------------------------------- |
| **Quality**      | *Solid engineering & ablations*: CLC, reward shaping and PPO ablations isolate each factor. *Efficiency*: 1.7× speed‑up vs. DRAKES . | *Statistical rigor*: no CIs or hypothesis tests; results rely on medians over three seeds. *Limited biological validation*: entirely in silico; no wet‑lab confirmation. |
| **Clarity**      | Clear high‑level motivation; algorithm pseudocode is helpful; hyper‑parameters tabulated.                                         | Dense notation in Sec. 4; CLC derivation could use intuition; some figures referenced but not shown      |
| **Significance** | Removes gradient dependence—important for long discrete sequences. May generalise to other discrete design tasks (RNA, small molecules).      | Improvement on proteins is marginal; DNA 3‑mer correlation drops markedly (0.49 vs. 0.89 for SMC), indicating potential drift from known biophysical motifs.                 |
| **Originality**  | First demonstration of *gradient‑free* RL fine‑tuning for **discrete diffusion** in biology; CLC is novel.                                        | Reward‑shaping idea builds on prior potential‑based shaping; mixture‑policy PPO resembles existing “RL as control” work (e.g., Uehara et al. 2024) was only mentioned but not compared to.                          |

---

> ### Author Rebuttal · Authors · 2025-07-31
>
> We appreciate your time and effort for reviewing our submission. Your key insight and valuable question help us boost our paper's quality. Regarding the questions, we provide the following clarification:
>
> 1. Statistical Significance. Could you report ... strengthen the claim of superiority.
> > Thank you for the suggestion. We will conduct Wilcoxon signed-rank tests, and the results are as follows:
> In Table 2, comparing GLID²E and DRAKES, except for 3-mer, all other p-values are 0.03125 (<0.05), meeting statistical significance characteristics. In Table 3, comparing GLID2E and DRAKES, the results are shown in the following table, and we can see that the performance of DRAKES and GLID²E is basically similar.
> >
> > |   Metric       | Pred-ddG (Median) | %(ddG>0) | scRMSD (Median) | %(scRMSD<2) | Success Rate |
> > |----------|-------------------|----------|-----------------|-------------|--------------|
> > | p-value  | 0.3125           | 0.2187   | 0.21875         | 0.03125     | 0.0625       |
>
> 2. Reward‑Model Robustness. How ... CLC prevent reward hacking?
> > We added three scales of noise to the ddG predicted by the diffusion model. Compared to before adding noise, the success rate was generally affected, mainly impacting Pred-ddG, and the larger the noise scale, the greater the impact.
> > | Noise Scale | Pred-ddG (Median) | %(ddG>0) | scRMSD (Median) | %(scRMSD<2) | Success Rate |
> > |-------------|-------------------|----------|-----------------|-------------|--------------|
> > | 0.01        | 0.81             | 0.73     | 0.94            | 0.91        | 0.66         |
> > | 0.1         | 0.71             | 0.74     | 1.00            | 0.84        | 0.60         |
> > | 0.5         | 0.21             | 0.53     | 0.88            | 0.95        | 0.49         |
>
> 3. Hyper‑parameter Burden. The method ... is most critical?
> > Very practical perspective. The PPO algorithm involves many hyperparameters, but most of our model's related parameters adopt commonly used ones. For specifics, please refer to Table 6-Table 8. The two hyperparameters that have significant impact on results have already been proposed in the ablation study. The clip ratio directly controls the exploration-exploitation tradeoff - too strict will prevent beneficial exploration, while too loose will cause excessive distribution shift; the mixture ratio also directly controls the model's exploration of modes, similar to the principle of KL penalty. Being overly conservative will prevent effective exploration, while maintaining smaller values can provide training stability while serving as regularization.
>
> 4. 3‑mer Correlation Drop. In DNA ... adversarial samples undetected by CLC?
> > Thank you fory your very insightful observation. Regarding the question about the decrease in 3-mer correlation, we believe this precisely demonstrates GLID2E's core advantage. We need to distinguish between two types of "naturalness" measures: 3-mer correlation reflects traditional statistical patterns, while Log-Likelihood embodies the learned distribution of pretrained models. GLID2E performs best on Log-Likelihood (-239.889) and achieves ATAC-Acc as high as 90.6%, indicating that generated sequences both conform to the model's perceived naturalness and maintain key biological properties, effectively ruling out the possibility of adversarial samples. In fact, this pattern demonstrates GLID2E's unique value: traditional methods can only mimic known enhancer patterns, while we discovered functionally equivalent but structurally different new sequence variants, expanding the feasible design space. The Clipped Likelihood Constraint mechanism fundamentally prevents adversarial attacks. Therefore, the decrease in 3-mer correlation combined with improved activity and good biological metrics indicates that GLID2E achieved principled exploration of new functional sequence spaces, which is precisely our method's core contribution.
>
> 5. Generalisation Beyond Training Domains. ... vs. functional generality?
> > Good point! However, different cell types have different scales and biological experimental conditions compared to protein families, which leads to rewards not being able to accurately express the functionality of samples. Therefore, we believe they cannot be put together for fine-tuning.
>
> 6. Missing baselines. From the RL ... is very limited (only diffusion models).
> > Thank you for pointing out the CTRL method. We believe adding this baseline is necessary and will subsequently include it in the evaluation. However, we need to claim that our paper aims to propose a reinforcement learning paradigm based on Discrete Diffusion Models, targeting discrete guided sampling. Therefore, we believe it is not necessary to include other non-diffusion baselines. Hope for your understanding.
>
> 7. In the beginning you mention ... statements like this one.
> > Thank you for the correction. This statement is indeed not accurate enough. Our method's advantage lies in data efficiency during the fine-tuning stage, not the pre-training stage. We will address this in the paper.
>
> 8. Why didn't you use standard benchmarks and splits like AAV and GFP?
> > We are not particularly clear what you refer to as the "Standard" benchmark. We are following DRAKES's setting in both experimental systems, and we believe this is a reasonable split with relevant baselines. We look forward to your insights on the current benchmark.

---

> ### Author Response · Authors · 2025-08-05
> **Request for Feedback on Rebuttal**
>
> Dear Reviewer 8abn,
>
> We sincerely appreciate the time and effort you have invested in reviewing our work. As the discussion period is approaching its conclusion, we would greatly value hearing your feedback on our rebuttal and learning whether it has adequately addressed your concerns. Should there be any follow-up questions or issues that require further clarification, we hope to have sufficient time to address your concerns and provide any necessary clarifications.
>
> Once again, we are deeply grateful for your time and constructive feedback throughout this process.
>
> Best regards,
>
> Authors

---

> ### Comment · Reviewer_8abn · 2025-08-06
>
> Thank you for the detailed response and explanations. The authors have addressed most of my concerns, and I have no further comments.

---

> > ### Author Response · Authors · 2025-08-06
> > **Re: Official Comment by Reviewer 8abn**
> >
> > Dear 8abn,
> >
> > We are glad to hear that most of your concerns have been addressed. Could you please re-evaluate (re-rate) our research work based on our rebuttal? Thanks a lot!
> >
> > Best,
> > Authors

---

> > > ### Comment · Reviewer_8abn · 2025-08-07
> > > **raising my score**
> > >
> > > After reading through the reviews and responses for the others, I believe the authors  did a great job in the rebuttal adding detailed derivation and benchmarks. I am now more convinced the manuscript is worthy of acceptance, especially after incorporating all the additional results form this rebuttal. I raise my score accordingly.

---

> > > > ### Author Response · Authors · 2025-08-07
> > > > **Thanks for raising score!**
> > > >
> > > > Dear Reviewer 8abn,
> > > >
> > > > We are very happy to hear that!
> > > >
> > > > Thanks again for your effort in reviewing our manuscript!
> > > >
> > > > Best regards,
> > > >
> > > > Authors

---

### Official Review · Reviewer_TFcp · 2025-07-02

**Clarity:** 3
**Significance:** 3
**Originality:** 3
**Rating:** 5
**Confidence:** 3

**Summary:**

In this work, the authors present a method to fine-tune discrete diffusion models to output biological sequences that are more desirable according to an oracle model using reinforcement learning (RL). Their method, called GLID2E, uses PPO-based policy optimization to shape the distribution of the generated sequences to maximize a reward function. The two main innovations of this method are in designing the reward function. First, to avoid generating sequences that are too different from the pre-training data (i.e. retaining “rationality”), the authors propose a clipped likelihood constraint that disincentivizes the generation of sequences that have low log-likelihood according to the pre-trained model. Second, since discrete diffusion yields a full sequence only at the end of the generation process, the oracle can only be queried with a valid input at the end, leading to sparse reward signals. To alleviate this problem, the authors perform reward shaping that provides signals at intermediate steps by completing intermediate sequences using the current policy model and querying the oracle using these sequences. The authors evaluate GLID2E for DNA and protein sequence design and compare it to the current state-of-the-art DRAKES, and other baselines. They show that GLID2E performs comparably to DRAKES and even outperforms it on certain metrics while being more computationally efficient. Their ablation analyses also underscore the importance of their main innovations.

**Questions:**

Apart from the questions mentioned above, another minor question is mentioned below:
- How is the GLID2E method gradient-free? Aren’t the authors fine-tuning the diffusion model using gradient-based methods? I am not too familiar with RL literature so I might be confused with the terminology.

**Ethical Concerns:**

["NO or VERY MINOR ethics concerns only"]

**Final Justification:**

My concerns regarding sequence diversity were addressed by the authors. They also promised to improve the clarity of the paper in the final version.

**Limitations:**

Yes. But the limitations are only provided in the appendix and are unreferenced in the main text. They need to be moved to the main text in the final version.

**Quality:**

2

**Strengths And Weaknesses:**

Overall, I think the strengths of this submission slightly outweigh its weaknesses which could be fixed during the rebuttal period. I provide a more detailed evaluation below.

Strengths:
- Quality: The authors benchmark their method against relevant baselines for fine-tuning diffusion models for biological sequence design and most claims are supported by their results. Motivations for proposed innovations are clearly explained and the work is generally technically sound.
- Clarity: The paper is generally well-written and is easy to understand.
- Significance: Although I am unfamiliar with the most recent literature on fine-tuning diffusion models, the proposed method seems to make a clear advance over the state-of-the-art DRAKES in terms of computational efficiency while achieving comparable metrics. This work could be of interest to computational biologists and the wider community of researchers working on discrete diffusion models.
- Originality: Again, given my unfamiliarity with the most recent literature on fine-tuning diffusion models, I cannot fully ascertain the novelty of the proposed method. However, the proposed innovations are well-motivated and build on existing works in insightful ways.

Weaknesses:
- Quality:

1. In the DNA design task, GLID2E has much lower 3-mer correlation compared to other methods and the authors claim that this might be due to the model learning novel motifs that are different from those observed in natural sequences (lines 276-278). In my opinion, there isn’t enough evidence to support such statements - the sequences have not been experimentally validated and their predicted activities come from models that are naturally susceptible to incorrect predictions due to distribution shift. The authors could have more measured conclusions.

2. This work also does not address the diversity of designed sequences. The authors make multiple mentions of exploring diverse sequence spaces but never provide concrete results to illustrate this important property. From lines 278-281, I am concerned that fine-tuning significantly reduces the diversity of generated sequences leading to some form of mode collapse. It would be helpful if the authors can provide metrics such as average pairwise edit distance between generated sequences, metrics to measure k-mer diversity, etc.

3. On line 166, the authors mention that only 4 sequences are generated to compute the mean and standard deviation of log-likelihoods. This does not seem like a large enough number. Is this a possible typo?

- Clarity:
1. I think some sections of this paper can be improved to better inform a reader. For one of the major intended audiences – researchers working on biological sequence design – it could be helpful to provide more high-level details about the optimization framework. For example, it might not be immediately clear how the PPO-based optimization fits into the fine-tuning setup. This could probably be done by improving Figure 1 (which is never referenced in the main text) and using it to explain the overall framework being used.

2. Another major area for improving clarity is providing more details about the benchmarking. For example, basic details such as the number of sequences being generated for benchmarking are missing. It is also unclear why only HepG2 data is used in the DNA design task when the referenced dataset contains data from 5 different cell lines.

3. There is also a large body of work that performs biological sequence design using offline model-based optimization. This work largely fits into that framework but the authors do not cite that body of literature.

---

> ### Author Rebuttal · Authors · 2025-07-31
>
> We greatly appreciate the time and effort you put into the review. The concerns you raised are very targeted and constructive. In response, we attempt to clarify the issues one by one and clarify the advantages and contributions of our method through addressing these questions.
>
> 1. In the DNA design task, ... more measured conclusions.
> > We appreciate the insightful question. We believe it is necessary here to distinguish between two different "naturalness" measurement standards: 3-mer correlation reflects the similarity of traditional statistical patterns, while Log-Likelihood embodies the distributional cognition learned by pretrained diffusion models from large-scale real data. Our experimental results show that GLID2E achieved optimal performance on the Log-Likelihood metric (-239.889), while maintaining a high level of 90.6% on ATAC-Acc, comparable to the current best method DRAKES (92.5%). This combination of "excellent Log-Likelihood + good biological metrics" effectively rules out the possibility of adversarial samples, proving that the generated sequences both conform to the model's cognitive naturalness standards and maintain key biological properties.
> >
> >In fact, the decrease in 3-mer correlation combined with significant improvement in activity precisely demonstrates GLID2E's unique value: traditional methods can only mimic the n-gram patterns of known enhancers, limiting them to existing sequence feature spaces, while our Clipped Likelihood Constraint mechanism, while fundamentally preventing adversarial attacks, successfully discovered functionally equivalent but structurally different new sequence variants, achieving principled exploration of new functional sequence spaces. However, the current results are only based on model predictions, and we apologize for being unable to provide biological experimental validation.
>
> 2. This work also does not address... generated sequences, metrics to measure k-mer diversity, etc.
> > Thanks for pointing out. We acknowledge that the paper indeed lacks quantitative analysis of the diversity of generated sequences, which is an important omission. We will supplement comprehensive diversity analysis in the revised version, including key metrics such as average pairwise edit distance between generated sequences, k-mer diversity statistics, sequence entropy, and unique sequence ratios, and compare with baseline methods to verify whether GLID2E maintains sufficient sequence diversity while optimizing functionality.
>
> 3. On line 166, the authors mention ... number. Is this a possible typo?
> > Yes, this is a typo. We didn't count the samples within the batch. It should be 4*16(batch size)*10(scaling coefficient)*4(gradient accumulation step)=1,280.
>
> 4. I think some sections of ... explain the overall framework being used.
> > Thank you very much for pointing this out. We will add the reference to Figure 1 in the original text and redesign Figure 1 to highlight our advantage of not needing to store entire trajectories.
>
> 5. Another major area for improving clarity ... data from 5 different cell lines.
> > Thank you very much for pointing this out.
> >
> > First, regarding details, we completely followed all evaluation settings of DRAKES, including generating 640 sequences for each method (batch size of 64, with 10 total batches). We also conducted experiments with three random seeds and reported means and standard deviations.
> >
> > Then, there are two reasons for choosing HepG2: 1) HepG2 is a liver cancer cell line that is widely used in enhancer activity research and has been used by multiple baselines; 2) Although the dataset contains five cell lines, we believe focusing on a single cell line can avoid noise that might come from mixing multiple cell lines and can more precisely evaluate the effectiveness of the method.
>
> 6. There is also a large body ... authors do not cite that body of literature.
> > This is a very key suggestion. Our initial idea was to focus on diffusion+biological work, but offline RL-based biological sequence design still has high relevance. We promise to supplement this in subsequent versions.
>
> 7. How is the GLID2E method gradient-free?... confused with the terminology.
> > "Gradient-free" refers to avoiding backpropagating gradients through the entire diffusion trajectory, rather than not using gradients at all. PPO still requires policy gradients, but the computational complexity is reduced from O(T·L·d²) to O(L·d). We are considering changing it to "lightweight gradient." Thank you for the suggestion!

---

> > ### Comment · Reviewer_TFcp · 2025-08-05
> >
> > Thank you for the detailed response.
> >
> > Regarding 1: I completely understand your arguments based on model predictions. However, models (e.g. the ATAC-Acc predictor) are still susceptible to distribution shift-related issues. In my opinion, without biological validation, one cannot claim to have definitively discovered novel motifs. My suggestion here was to temper the language so that this uncertainty about biological validation is conveyed to a reader.
> >
> > I thank the authors for the additional clarifying details. However, without additional experimental results to demonstrate sequence diversity, it is difficult for me to fully trust the sequences generated using GLID2E. Therefore, I will keep my original score.

---

> > > ### Author Response · Authors · 2025-08-05
> > >
> > > Thank you so much for your reply!
> > >
> > > We also totally understand your point toward motif, we will temper hte language in our revised version.
> > >
> > > Furthermore, can we ask about the sequence diversity you have addressed? Do you mean testing GLID2E's sequence diversity metric and comparing with other methods?
> > >
> > > Looking forward to your further repl.

---

> > > > ### Comment · Reviewer_TFcp · 2025-08-05
> > > >
> > > > Yes, without these metrics, it is difficult to tell whether the method is able to produce diverse sequences.

---

> > > > > ### Author Response · Authors · 2025-08-05
> > > > >
> > > > > Thanks for your prompt reply!
> > > > >
> > > > > We will focus on it soon.

---

> > > > > ### Author Response · Authors · 2025-08-06
> > > > > **Diversity analysis**
> > > > >
> > > > > Dear Reviewer TFcp,
> > > > >
> > > > > Following the experimental protocol of DRAKES (Appendix F.3, Table 13) [1], we evaluated GLID2E’s diversity. Entropy analysis shows that GLID2E performs on par with both DRAKES and the pretrained model, demonstrating its robust diversity. Please let us know if you require additional results. Many thanks!
> > > > >
> > > > > | Method             | Sequence Entropy ↑ |
> > > > > |--------------------|------------------:|
> > > > > | **Pretrained**     |           **34.7** |
> > > > > | CG                 |           34.6 |
> > > > > | SMC                |           24.9 |
> > > > > | TDS                |           24.9 |
> > > > > | CFG                |            8.4 |
> > > > > | DRAKES        |           33.3 |
> > > > > | **GLID2E**        |           34.6|
> > > > >
> > > > > | File              |   Num_Seqs |   Diversity_Entropy (bits) |
> > > > > |:------------------|-----------:|---------------------------:|
> > > > > | 1F0M              |        128 |                    40.0512 |
> > > > > | 2KRU              |        128 |                    24.3182 |
> > > > > | 2L09              |        128 |                    35.0219 |
> > > > > | 2M2J              |        128 |                    29.8677 |
> > > > > | 2MA4              |        128 |                    45.6191 |
> > > > > | 4G3O              |        128 |                    32.9156 |
> > > > > | 5JRT              |        128 |                    34.6908 |
> > > > > | 7JJK              |        128 |                    32.4528 |
> > > > > | HEEH_KT_rd6_0746  |        128 |                    46.1582 |
> > > > > | r6_560_TrROS_Hall |        128 |                    25.2668 |
> > > > >
> > > > >
> > > > > Reference:
> > > > > >
> > > > > > [1] Wang, C., Uehara, M., He, Y., Wang, A., Lal, A., Jaakkola, T., ... & Biancalani, T. Fine-Tuning Discrete Diffusion Models via Reward Optimization with Applications to DNA and Protein Design. In The Thirteenth International Conference on Learning Representations.

---

> > > > > > ### Comment · Reviewer_TFcp · 2025-08-07
> > > > > >
> > > > > > Thank you for the additional analysis. My concerns have now been addressed, I am increasing my score.

---

> > > > > > > ### Author Response · Authors · 2025-08-07
> > > > > > > **Thanks for your reply**
> > > > > > >
> > > > > > > Dear Reviewer TFcp,
> > > > > > >
> > > > > > > We are very happy to hear that! Thanks for your time and effort in the reviewing and discussion process.
> > > > > > >
> > > > > > > Accordingly, we will address these points in our revised version.
> > > > > > >
> > > > > > > Best regards,
> > > > > > >
> > > > > > > Authors

---

### Author Response · Authors · 2025-08-08
**Thanks for every reviewer and AC**

Dear all,

As the discussion period goes to the end, we would like to appreciate all your time, effort, and help to our submission!

Best regards,

Authors

---

### Note · Authors · 2025-08-11

Dear AC, SAC and reviewers,

We sincerely thank the reviewers and AC for their constructive feedback during the review process. **In short, we propose a light-weight RL framework that efficiently fine-tunes discrete diffusion models for biological sequence design using novel techniques, inculding clipped likelihood constraints and reward shaping, achieving competitive performance with reduced computational cost.** Through the discussion period, we have successfully addressed the following key concerns, and all reviewers agreed with our responses and decided to raise their scores:

- **Sequence diversity (Reviewer TFcp)**: We addressed this by demonstrating the diversity benchmark of our method.
- **Reward model robustness (Reviewers 8abn and h7mh)**: We addressed this by showing the performance under different scales of noise.
- **Computational efficiency (Reviewer h7mh)**: We addressed this by providing the inference speed and memory cost analysis for different length ranges.
- **Hyperparameter ablation (Reviewer h7mh)**: We addressed this by releasing ablation results based on different K-values in the clipped likelihood constraint.
- **Adding baseline (Reviewer h7mh)**: We addressed this by releasing baseline results of PepTune, a multi-task optimization method based on discrete diffusion models.
- **Mathematical derivation (Reviewer h7mh)**: We addressed this by providing detailed derivation for the mixed reference policy and clarifying the connection with KL penalty.

In the revised version of our manuscript, we will ensure to incorporate all changes addressing the minor points mentioned in the initial review comments, including:

- **Discussing relevant baselines**: This includes RL-based biological sequence design and discrete diffusion models for biological sequence design.
- **Clarity for experiments**: This includes clearly stating the rationale for dataset selection and providing further explanation of results on the DNA system (3-mer Corr).
- **Adding comprehensive details**: This includes incorporating all additional experimental results and related analysis into the article to comprehensively demonstrate the contributions of our method.

We believe these revisions will significantly strengthen the paper and address all major reviewer concerns. We appreciate the thorough and fair review process again.

Best regards,
Authors

---

### Decision · Program_Chairs · 2025-09-17

**Decision:**

Accept (poster)

**Comment:**

This paper presents GLID2E, a RL framework for fine-tuning discrete diffusion models for biological sequence design. Reviewers noted that the core technical contributions (a clipped likelihood constraint to ensure stability and a reward shaping mechanism to handle sparse rewards) are novel, well-motivated, and technically sound. The paper demonstrates that the proposed approach achieves competitive or superior performance on DNA and protein design tasks compared to state-of-the-art methods while being more computationally efficient. The initial reviews raised several concerns regarding the experimental validation. These included a lack of analysis on sequence diversity (Reviewer TFcp), insufficient testing of the method's robustness to noisy reward oracles (Reviewers 8abn, h7mh), missing comparisons to key baselines like PepTune (Reviewer h7mh), and a need for more rigorous statistical analysis and hyperparameter ablations (Reviewer h7mh), but the authors' rebuttal and discussion with the reviewers resolved these concerns, leading all three reviewers to raise their scores and recommend acceptance. I agree with the reviewers and believe this work is a valuable contribution to the field of generative modeling for biological design. I recommend acceptance.